# Additive MIL: Intrinsically Interpretable Multiple Instance Learning for Pathology

**Syed Ashar Javed**
PathAI Inc
ashar.javed@pathai.com

**Dinkar Juyal**
PathAI Inc
dinkar.juyal@pathai.com

**Harshith Padigela**
PathAI Inc
harshith.padigela@pathai.com

**Amaro Taylor-Weiner**
PathAI Inc
amaro.taylor@pathai.com

**Limin Yu**
PathAI Inc
limin.yu@pathai.com

**Aaditya Prakash**
PathAI Inc
adi.prakash.ml@gmail.com

## Abstract

Multiple Instance Learning (MIL) has been widely applied in pathology towards solving critical problems such as automating cancer diagnosis and grading, predicting patient prognosis, and therapy response. Deploying these models in a clinical setting requires careful inspection of these black boxes during development and deployment to identify failures and maintain physician trust. In this work, we propose a simple formulation of MIL models, which enables interpretability while maintaining similar predictive performance. Our Additive MIL models enable spatial credit assignment such that the contribution of each region in the image can be exactly computed and visualized. We show that our spatial credit assignment coincides with regions used by pathologists during diagnosis and improves upon classical attention heatmaps from attention MIL models. We show that any existing MIL model can be made additive with a simple change in function composition. We also show how these models can debug model failures, identify spurious features, and highlight class-wise regions of interest, enabling their use in high-stakes environments such as clinical decision-making.

## 1 Introduction

Histopathology is the study and diagnosis of disease by microscopic inspection of tissue. Histologic examination of tissue samples plays a key role in both clinical diagnosis and drug development. It is regarded as medicine's ground truth for various diseases and is important in evaluating disease severity, measuring treatment effects, and biomarker scoring [42]. A differentiating feature of digitized tissue slides or whole slide images (WSI) is their extremely large size, often billions of pixels per image. In addition to being large, WSIs are extremely information dense, with each image containing thousands of cells and detailed tissue regions that make manual analysis of these images challenging. This information richness makes pathology an excellent application for machine learning, and indeed there has been tremendous progress in recent years in applying machine learning to pathology data [43, 9, 19, 46, 12, 7, 14, 13].

The most important applications of ML in digital pathology involve predicting patient's clinical characteristics from a WSI image. Models need to be able to make predictions about the entire slide

involving all the patient tissue available; we refer to these predictions as "slide-level". To overcome the challenges presented by the size of these images, previous methods have used smaller hand engineered representations, built from biological primitives in tissue such as cellular composition and structures [12]. Another common way to overcome the challenges presented by the size of WSIs is to break the slide into thousands of small patches, train a model with these patches to predict the slide-label, and then use a secondary model to learn an aggregation function from patch representations to slide-level label [9]. Both methods are not trained in an end-to-end manner and suffer from sub-optimal performance. The second method also suffers from an incorrect assumption that each patch from a slide has the same label as the overall slide [17].

Multiple Instance Learning [30] is a weakly supervised learning technique which attempts to learn a mapping from a set of instances (called a bag) to a single label associated with the whole bag. MIL can be applied to pathology by treating patches from slides as instances which form a bag and a slide-level label is associated with each bag to learn a bag predictor. This circumvents the need to collect patch-level labels and allows end-to-end training from a WSI. The MIL assumption that at least one patch among the set of patches is associated with the target label works well for many biological problems. For example, the MIL assumption holds for the task of cancer diagnosis; a sufficiently large bag of instances or patches from a cancerous slide will contain at least one cancerous patch whereas a benign slide will never contain a cancerous patch. In recent years, attention based pooling of patches has been shown to be successful for MIL problems [17]. Using neural networks with attention MIL has become the standard for end-to-end pathology models as it provides a powerful, yet efficient gradient based method to learn a slide-to-label mapping. In addition to superior performance, these models encode some level of spatial interpretability within the model through visualization of highly attended regions.

The sensitive nature of the medical imaging domain requires deployed machine learning models to be interpretable for multiple reasons. First, it is critical that models do not learn spurious shortcuts over true signal [11, 35] and can be debugged if such failure modes exist. Interpretability and explainability methods have been shown to help identify some of these data and model deficiencies [15, 27, 4, 39]. Secondly, for algorithms in medical decision-making, accountability and rigorous validation precedes adoption [26, 18]. Interpretable models can be easier to validate and thus build trust. Specifically, users can verify that model predictions are generated using biologically concordant features that are supported by scientific evidence and are similar to the those identified by human experts. Thirdly, use-cases involving a human expert such as decision-support require the algorithm to give a visual cue which highlights the regions to be examined more carefully. In these applications, a predicted score is insufficient and needs to be complemented with a highlighted visual region associated with the model's prediction [8].

For ML models in pathology, spatial credit assignment can be defined as attributing model predictions to specific spatial regions in the slide. Various post-hoc interpretability techniques like gradient based methods [33] and Local Interpretable Model-agnostic Explanation (LIME) [44] have been used to this end. However, gradient based methods which try to construct model-dependent saliency maps are often insensitive to the model or the data [20, 2, 5]. This makes these post-hoc methods unreliable for spatial attribution as they provide poor localization and do not reflect the model's predictions [15]. Model-agnostic methods like Shapley values or LIME involve intractable computations for large image data and thus need approximations like locally fitting explanations to model predictions, which can lead to incorrect attribution [32]. Applying Attention MIL [17] in weakly supervised problems in pathology leads to learning of the attention scores for each patch. These scores can be used as a proxy for patch importance, thus helping in spatial credit assignment. This way of interpreting MIL models has been used commonly in the literature to create spatial heatmaps, image overlays that indicate credit assignment, for free without applying any post-hoc technique [47, 27, 24, 36, 23]. The attention values that scale patch feature representations have a non-linear relationship to the final prediction, making their visual interpretation inexact and incomplete. We discuss these issues in detail in the next section.

To resolve these issues, we propose a simple formulation of MIL which induces intrinsically interpretable heatmaps. We refer to this model as Additive MIL. It allows for exact decomposition of a model prediction in terms of spatial regions of the input. These models are inspired by Generalized Additive Models (GAMs) [16] and Neural Additive Models (NAMs) [3], but instead of being applied to arbitrary features, they are grounded as patch instances in the MIL formulation which allows exact credit assignment for each patch in a bag. Specifically, we achieve this by constraining the space

of *predictor functions* (the classification or regression head at the final layer) in the MIL setup to be additive in terms of instances. Therefore, the exact contribution of each patch or instance in a bag can be traced back from the final predictions. We show that these additive scores reflect the true marginal contribution of each patch to a prediction and can be visualized as a heatmap on a slide for various applications like model debugging, validating model performance, and identifying spurious features. We also show that these benefits are achieved without any loss of predictive performance even though the predictor function is now fixed to be additive. This is critical as the accuracy-interpretability trade-off has been an active area of research and has deep implications for applications in medical imaging. Trading off performance for interpretability might make sense for improving validation and in turn adoption of ML tools, however it raises ethical questions about deploying sub-optimal models [34, 25, 10]. Furthermore, since our work is orthogonal to previous advancements in MIL modeling, we show that previous methods can be made additive by a simple switching of function composition at the last layer of the model, making it applicable to all MIL models where instance-level credit assignment is important.

## 2 MIL Models & Interpretability

### 2.1 Interpretability in Attention MIL

An attention MIL model [17] can be seen as a 3-part model consisting of a featurizer ($f$), typically a deep CNN, an attention module ($m$) which induces a soft attention over the $N$ patches and is used to scale each patch feature, and a predictor ($p$) which takes the attended patch representations, aggregates them using a permutation invariant function like sum pooling over the $N$ patches, and then outputs a prediction. This MIL model $g(x)$ is given as:

$$g(x) = (p \circ m \circ f)(x) \tag{1}$$

$$m_i(x) = \alpha_i f(x_i) \quad \text{where} \quad \alpha_i = softmax_i(\psi_m(x)) \tag{2}$$

$$p(x) = \psi_p(\sum_{i=1}^{N} m_i(x)) \tag{3}$$

where $\psi_m$ and $\psi_p$ are MLPs with non-linear activation functions.

The attention scores $\alpha_i$ learned by the model can be treated as patch importance scores and have been used previously to interpret MIL models [47, 27, 36, 23]. However, there are four issues in doing spatial attribution using these attention scores. For example, consider the task of classifying a slide into benign, suspicious or malignant:

(a) Since the attention weights are used to scale the patch features used for the prediction task, a high attention weight only means that the patch might be needed for the prediction downstream. Therefore, a high attention score for a patch can be a necessary but not sufficient condition for attributing a prediction to that patch. Similarly, patches with low attention can be important for the downstream prediction since the attention scores are related non-linearly to the final classification or regression layer. For example, in a malignant slide, non-tumor regions might get highlighted by the attention scores since they need to be represented at the final classification layer to provide discriminative signal. However, this does not imply malignant prediction should be attributed to non-malignant regions, nor that these regions would be useful to guide a human expert.

(b) The patch's contribution to the final prediction can be either positive (excitatory) or negative (inhibitory), however attention scores do not distinguish between the two. A patch might be providing strong negative evidence for a class but will be highlighted in the same way as a positive patch. For example, benign mimics of cancer are regions which visually look like cancer, but are normal benign tissue [40]. These regions are useful for the model to provide negative evidence for the presence of cancer and thus might have high attention scores. While attending to these regions may be useful to the model, they may complicate human interpretation of resulting heatmaps.

(c) Attention scores do not provide any information about the class-wise importance of a patch, but only that a patch was weighted by a certain magnitude for generating the prediction. In the case of multiclass classification, this becomes problematic as a high attention scores on a patch can mean that it might be useful for any of the multiple classes. Different regions in the slide might be contributing to different classes which are indistinguishable in an attention heatmap. For

example, if a patch has high attention weight for benign-suspicious-malignant classification, it can be interpreted as being important for any one or more of the classes. This makes the attention scores ineffective for verifying the role of individual patches for a slide-level prediction.

(d) Using attention scores to assess patch contribution ignores patch interactions at the classification stage. For example, two different tumor patches might have moderate attention scores, but when taken together in the final classification layer, they might jointly provide strong and sufficient information for the slide being malignant. Thus, computing marginal patch contributions for a bag needs to be done at the classification layer and not the attention layer since attention scores do not capture patch interactions and thus can under or over estimate contributions to the final prediction.

These difficulties in interpreting attention MIL heatmaps motivate the formulation of a traceable predictor function, where model predictions can be exactly specified in terms of patch contributions (both positive and negative) for each class.

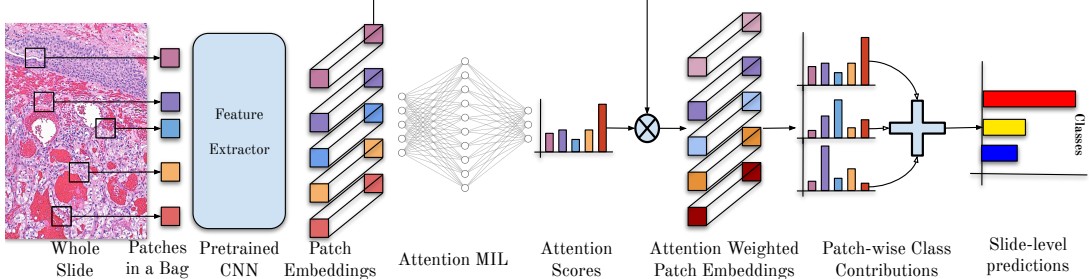

Figure 1: Additive MIL Model

## 2.2  Additive MIL Models

We first define the desiderata for a visual interpretability method for MIL models:

1. The method should be intrinsic to the model and not be a post-hoc method. This prevents incorrect assumptions about the model and does not require post-hoc modeling. It also prevents many pitfalls of traditional saliency methods [20, 2].

2. Attribution in the MIL setup should be in terms of instances only. For pathology, this means that the prediction should be attributed to individual patches. This constraint enables expression of bag predictions in terms of marginal instance contributions.

3. Should reflect model's sensitivity and invariances by reliably mirroring its functioning [22].

4. Should distinguish between excitatory and inhibitory patch contributions. Should also provide per-class contributions for classification problems.

To enable the desired instance-level credit assignment in MIL, we re-frame the final predictor to be an additive function of individual instances. This translates to a simple switching of the function composition in Equation 3:

$$p_{Additive}(x) = \sum_{i=1}^{N} \psi_p(m_i(x)) \tag{4}$$

Making this change results in the final predictor only being able to implement patch-additive functions on top of arbitrarily complex patch representations. This provides both complexity of the learned representations as well as a traceable patch contribution for a given prediction which solves the spatial credit assignment problem. $\psi_p(m_i(x))$ is the class-wise contribution for patch $i$ in the bag. At inference, $\psi$ produces a $R^{C \times N}$ for a classification problem where $C$ is the number of classes and $N$ is the number of patches in a bag. Thus, we get a class-wise score for each patch, which when summed gives the final logits for the prediction problem. These scores can be visualized by constructing a heatmap from the visual representation of patch-wise contributions for each class. The sign of the patch contribution decides whether the patch is excitatory or inhibitory towards each class

since positive values add to the final logit while negative values bring down the final class logit. In the next section, we prove that the instance contribution obtained from an Additive MIL model is exactly equivalent to the actual marginal contribution of that patch to a model's prediction.

## 2.3 Proof of equivalence between Additive MIL and Shapley Values

We highlight the spatial credit assignment properties of an Additive MIL model by proving its equivalence to Shapley values [37]. Shapley value is game theoretic concept used for calculating the optimal marginal contribution of each player in a n-player coalition game with a given total payoff. In machine learning, it's used to interpret models predictions by decomposing them in terms of their marginal feature contributions. There have been various applications of using Shapley values for feature credit assignment in the interpretability literature [28, 31, 39, 38, 1]. However for most practical models, the computational complexity grows exponentially, thus requiring approximations [41]. Here, we show that our additive MIL formulation is equivalent to Shapley values without any approximation.

The Additive MIL model, $g$ is defined for a MIL problem with instances $x_i$ as input:

$$g(x) = \sum_{n=1}^{N} \psi_p(\alpha_i f(x_i)) \tag{5}$$

where $\alpha_i$ is the attention weight for the $i^{th}$ instance, $f$ is the function encoding each instance into a feature representation, and $\psi_p$ is predictor function which maps instance representation to the model output (e.g. logits for classification models).

**Theorem 1:** *The marginal instance contribution from an Additive* MIL *model,* $g(x_i)$ *is proportional to the Shapley value of that instance,* $\phi_i$.

$$g(x_i) \propto \phi_i(V, x) = \sum_{S \subseteq F \setminus i} \frac{|S|!(|F| - |S| - 1|!)}{|F!|} V_{S \cup i}(x_{S \cup i}) - V_S(x_S) \tag{6}$$

*Consequence:* Additive MIL scores ensure optimal credit assignment across instances of an MIL bag. Thus each bag-level prediction in MIL can be exactly decomposed into marginal instance contributions given by Additive MIL scores and provide model interpretability.

*Proof:* The proof is in Appendix A (see Supplementary Material)

## 2.4 Features of Additive MIL Models

**Exact marginal patch contribution towards a prediction**. Additive MIL models provide exact patch contribution scores which are additively related to the prediction. This additive coupling of the model and the interpretability method makes the spatial scores precisely mirror the invariances and the sensitivities of the model, thus making them intrinsically interpretable.

**Class-wise contributions**. Additive MIL models allow decomposing the patch contributions and attributing them to individual classes in a classification problem. This allows us to not just assign the prediction to a region, but to also see which class it contributes to specifically. This is helpful in cases where signal for multiple classes exist within the same slide.

**Distinction between excitatory and inhibitory contributions**. Additive MIL models allow for both positive and negative contributions from a patch. This can help distinguish between areas which are important because they provide evidence for the prediction and those which provide evidence against.

## 3 Experiments & Results

### 3.1 Experimental Setup & Datasets

We perform various experiments to show the benefits of using Additive MIL models for interpretability in pathology problems. Concretely, we show the following results:

- Additive MIL models provide intrinsic spatial interpretability without any loss of predictive performance as compared to more expressive, non-additive models.

- Any pooling-based MIL model can be made additive by reformulating the predictor function and leads to predictive results similar to the original model.

- Additive MIL heatmaps yield better alignment with region-annotations from an expert pathologist than Attention MIL heatmaps.

- Additive MIL heatmaps provide more granular information like class-wise spatial assignment and excitatory and inhibitory patches which is missing in attention heatmaps. This can be useful for applications like model debugging.

We consider 3 different datasets and 2 different problems for our experiments. The first problem is the prediction of cancer subtypes in non-small cell lung carcinoma (NSCLC) and renal cell carcinoma (RCC), both of which use the TCGA dataset [45]. The second problem is the detection of metastasis in breast cancer using the Camelyon16 dataset [6]. TCGA RCC contains 966 whole slide images (WSIs) with three histologic subtypes - KICH (chromophobe RCC), KIRC (clear cell RCC) and KIRP (papillary RCC). We extract $768k$ patches from this dataset which translates to an average of 795 patches per slide and $16k$ total bags. TCGA NSCLC has 1002 WSIs, with 538 slides belonging to subtype LUAD (Lung Adenocarcinoma) and 464 to LUSC (Lung Squamous Cell Carcinoma). We extract 1.465 million patches from this dataset which translates to an average of 1462 patches per slide and $30.5k$ total bags. Camelyon16 contains 267 WSIs for training and 129 for testing with a total of 159 malignant slides and 237 benign slides. We extract $510k$ patches from this dataset which translates to an average of 1286 patches per slide and $10.6k$ total bags. These numbers point to the diversity in the dataset size in terms of number of slides, number of bags, and the label imbalance.

### 3.2 Implementation Details

Both TCGA datasets were split into 60/15/25 (train/val/test) as done previously [36] while ensuring no data leakage at a case level. For Camelyon16, we use the original splits provided with the dataset. For training the models, a bag size of 48-1600 patches and batch size of 16-64 was experimented with and the best one chosen using cross-validation. The patches were sampled from non-background regions for all datasets at a resolution of 1 microns per pixel without any overlap between adjacent patches. An ImageNet pre-trained Shufflenet [29] was used as the feature extractor and the entire model was trained with ADAM optimizer [21] and a learning rate of 1e-4. For inference, multiple bag predictions were aggregated using a majority vote to get the final slide-level prediction. AUROC (area under the receiver operating curve) scores were generated using the proportion of bags predicting the majority label as the class assignment probability. For TCGA-RCC, we compute macro average of 1-vs-rest AUROC across the 3 classes. The attention scores were obtained by directly taking the raw outputs for each patch from the attention module. For additive patch contributions, the patch-wise class contributions were taken and converted to a bounded patch contribution value using a sigmoid function. This yielded excitatory scores in the range of $0.5 - 1$ and inhibitory scores in the range of $0 - 0.5$. Both the attention and additive patch-wise scores were used for generating a heatmap as an overlay on the slide with Attention MIL heatmaps having a single value per patch and Additive MIL heatmaps having $C$ values per patch where $C$ is the number of classes. All training and inference runs were done on Quadro RTX 8000, and it takes 3 to 4 hours to train the model with four GPUs.

### 3.3 Experimental Results

#### 3.3.1 Predictive Performance of Additive MIL Models & Applicability to Previous Methods

We compare Additive MIL models with existing techniques in terms of predictive performance on 3 different datasets. We implement a mean-pooling based MIL baseline without any attention, the standard Attention MIL model (ABMIL) and a transformer based MIL model, TransMIL which is the state-of-the-art on these three datasets (for comparison of TransMIL with other methods, refer [36]). Table 1 shows how Additive MIL models achieve comparable or superior performance to the standard Attention MIL model. In the case of improved performance, we hypothesize that the additive constraint regularizes the model and limits overfitting in comparison to previous approaches. This is particularly relevant to pathology datasets that often have less than one thousand slides. The results in the table also demonstrate how previous techniques like TransMIL can be made additive by switching the function composition of the classifier layer as done in Equation 3 and 4. This property is general, thus any high performing MIL method can be converted to an Additive MIL model. Implementing the

| Method | Camelyon16 | | TCGA NSCLC | | TCGA RCC | |
|---|---|---|---|---|---|---|
| | Accuracy | AUC | Accuracy | AUC | Accuracy | AUC |
| Mean Pooling MIL | 0.751 | 0.707 | 0.830 | 0.925 | 0.918 | 0.980 |
| Mean Pooling MIL + Additive | 0.734 | 0.687 | 0.866 | 0.924 | 0.902 | 0.974 |
| Attention MIL [ABMIL] [17] | 0.773 | 0.750 | 0.883 | 0.946 | 0.878 | 0.978 |
| Attention MIL + Additive | 0.830 | 0.846 | 0.886 | 0.941 | 0.915 | 0.983 |
| TransMIL [36] | 0.805 | 0.775 | 0.878 | 0.932 | 0.915 | 0.983 |
| TransMIL + Additive | 0.805 | 0.844 | 0.895 | 0.934 | 0.911 | 0.986 |

Table 1: Comparison of predictive performance on Camelyon16, TCGA NSCLC & RCC datasets.

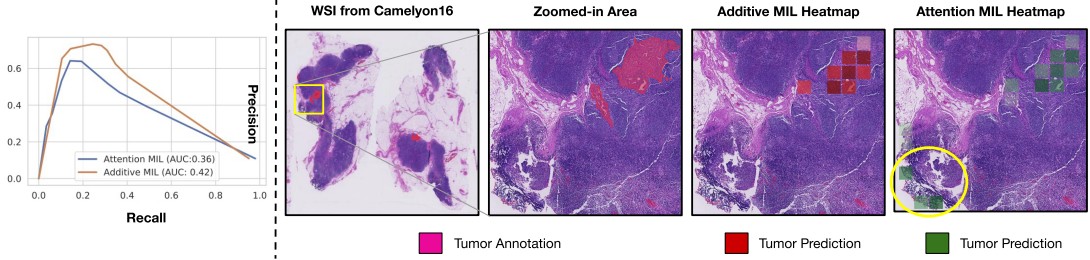

Figure 2: Comparison of Additive & Attention MIL heatmaps at detecting annotated cancer regions in Camelyon16. Attention MIL heatmaps have lower precision and detect false-positives as highlighted in the yellow circle.

additive formulation gives nearly all the benefits of modeling complexity from previous methods, while enabling spatial interpretability without any loss of predictive performance.

### 3.3.2 Region-level Alignment of Additive Heatmaps with Expert Pathologist

Here we compare heatmaps obtained through Additive MIL models with attention heatmaps and evaluate both against region-level annotations from an expert pathologist. We use the Camelyon16 dataset where the goal is to classify the slide as benign or malignant. Since the cancer foci are very localized and often occupy less than 1% of the slide, the task of generating localized cancer heatmaps in a weakly supervised setup is very challenging. We obtained exhaustive segmentation annotations for cancer regions from a board-certified pathologist on the Camleyon16 test set. We trained an Additive MIL model and generated both the traditional attention heatmaps using the patch-level attention scores and Additive MIL heatmaps using the patch contributions. We compare the patch level precision-recall curves at different thresholds of the heatmap. Note that this comparison controls for model performance as both heatmaps are generated from the same model. Figure 2 shows this comparison. At low thresholds, nearly all patches are highlighted and we see a high recall and low precision for both methods. As we increase this threshold we see higher precision and lower recall. The Additive MIL heatmaps (AUPRC 0.42) highlighted cancer regions more precisely and sensitively than traditional attention heatmaps (AUPRC 0.36), which detect more false-positives (Figure 2). If we choose the best operating point of both the curves, we find that the best F1 score for the attention heatmap is 0.43 as compared to 0.47 from the Additive heatmap. These experiments demonstrate the superior performance of Additive MIL heatmaps in localizing areas of interest.

### 3.3.3 Faithful Representation of Patch-level Contributions to Slide-level Predictions

Attention heatmaps are often used to signal regions of interest in a slide, however as argued in Section 2.1, it is not straightforward to draw conclusion regarding the importance and contribution of 'attended' areas towards the model prediction. Additive MIL guarantees that each patch's contribution is linear and thus faithfully represents its marginal contribution toward the slide-level prediction. This property is shown in Figure 3, where the linear relationship of Additive MIL (top row) is clear. In

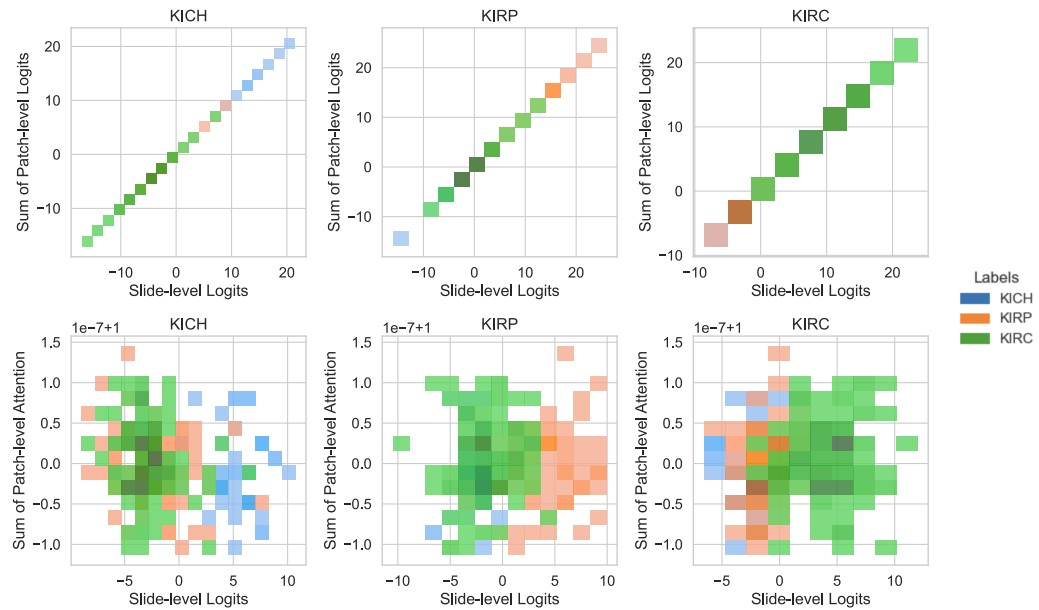

Figure 3: Figure shows the alignment between the slide-level predicted logits and patch contributions from the Additive and the Attention models on TCGA RCC. Top-row: Y-axis shows the sum of patch contribution in a bag for **Additive**. Bottom-row: Y-axis shows the median score from top-10% patches in a bag for **Attention**. The columns represent the slide-level logits for each class. The colors represent the ground-truth. Additive contributions are linear while Attention is not.

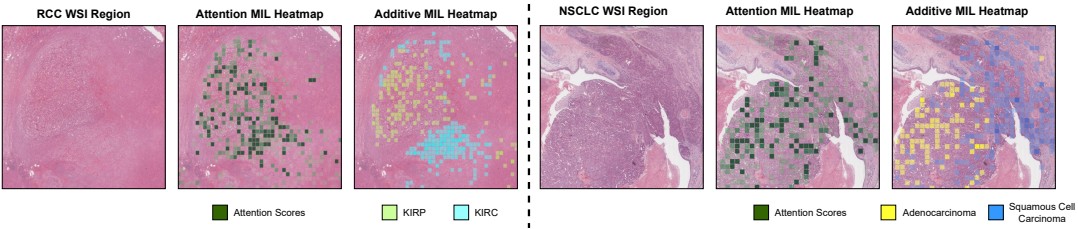

Figure 4: Even in cases where attention heatmap identifies the same region as an Additive MIL heatmap, the ability of the latter to show granular class-wise patch contributions can lead to better interpretation. The left 3 column (RCC) shows the Additive MIL heatmaps: KIRC regions in cyan and KIRP in green. The right 3 columns show Adenocarcinoma regions in yellow and Squamous cell carcinoma in blue. Attention heatmaps are incapable of visualizing such class-level information.

contrast, when considering the attention scores of the most attended patches (top 10% of patches), there is no relationship with the final predictions (shown in the bottom row).

### 3.3.4 Qualitative Assessment of Multi-Class & Excitatory-Inhibitory Heatmaps

We highlight the benefits of having class-wise excitatory-inhibitory contributions for each spatial region in a slide. Figure 4 shows zoomed-in regions of two slides from TCGA RCC and NSCLC. In these examples, the attention heatmaps do highlight tissue regions predictive of the cancer subtype but don't provide information about the association of patches to classes. In contrast, the Additive MIL heatmaps show precisely how each patch contributes to each class, and in turn the final prediction. Figure 5 shows the information about excitatory and inhibitory patches for different classes. The Additive MIL heatmaps for each class are visualized by the same colorbar where red denotes excitatory patches and blue denotes inhibitory ones. The RCC WSI is labeled as KIRC, but the selected region contains two subtypes, namely KIRC and small regions of KIRP, as evident from the raw slide. The Additive MIL heatmaps accurately show bottom right region being excitatory for KIRC, but inhibitory for the other two whereas the top left region is only excitatory for KIRP and inhibitory for two

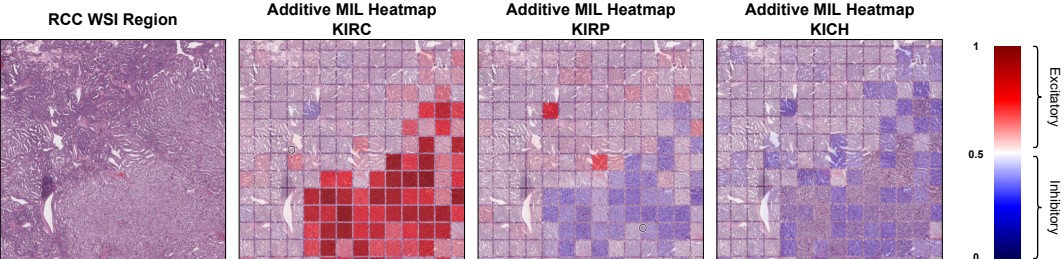

Figure 5: Additive MIL heatmaps provide excitatory and inhibitory patch contributions for each class which can be analyzed to understand how each region is voting for or against each class. Red denotes excitatory while blue denotes inhibitory contributions for each class. In the same RCC slide, regions containing morphological signal for multiple subtypes can be seen and corroborated using the excitatory-inhibitory patches from different regions, thus helping in model evaluation.

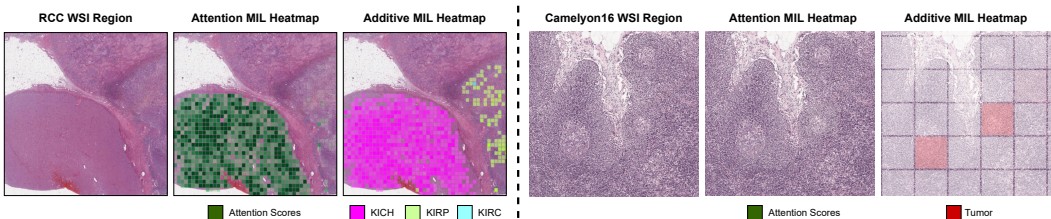

Figure 6: Left 3 columns show the case of model mis-predicting a KIRP slide as KICH. Attention heatmaps show a region of adrenal gland on the left being attended. Additive MIL heatmaps are able to exactly show how adrenal glands being rare, are being confused for KICH regions even though the model correctly identifies the KIRP regions on the right side. The right 3 columns show a case from Camelyon16 where the model is mis-predicting a benign slide as malignant. The attention heatmap offers no information, however, Additive MIL heatmap highlights areas of germinal center as the source of this false positive prediction (in red). This pattern for false positive prediction is found in multiple other slides and can enable us to go from interpretation to debugging.

other two. All patches are correctly inhibitory for KICH. Such granularity in heatmaps is helpful in understanding how the model arrives at a prediction and can prove to be useful for practitioners building the models as well as physicians using them.

### 3.3.5 Model Debugging Using Additive MIL Heatmaps

The ability of Additive MIL models to perfectly reflect model predictions at a patch-level can be useful in model debugging. Here, we show examples of the cases where Additive MIL heatmaps identify reasons for model failures during our experiments. Figure 6 shows two such cases. These heatmaps not only provide interpretability to MIL models, but can also aid in validating specific hypothesis during model debugging.

### 3.3.6 Limitations

Since the interpretation of Additive MIL models is based on model predictions that are reformulated to be interpretable, the interpretability method is inherently coupled with the model. This is desirable since the heatmaps now exactly track model predictions, but this coupling also potentially limits the flexibility of the models and heatmaps. For example, since the patch contributions are tied to the model, one can only generate heatmaps with patches at the same resolution as what the model was trained on. Another limitation is the reduction of model expressivity introduced by the additive constraint. In this study, we did not find a practical example of this limitation, however, it may exist in other datasets.

# 4   Conclusion & Broader Impact

We propose a simple reformulation of the popular Attention MIL setup for pathology that makes models intrinsically interpretable through an additive function. Our approach enables exact spatial credit assignment where the final prediction of the model can be attributed to individual contributions of each patch in a pathology slide. These models provide spatial interpretability without any loss of predictive performance and can be used for various applications like model debugging and highlighting regions-of-interest in a decision-support setting. This high fidelity interpretability will be critical in building trust for these models when deployed in medical decision-making.

## Acknowledgements

The authors would like to thank Francesco Rubbo for valuable feedback on the paper draft. The authors would also like to thank Aryan Pedawi and Harsha Pokkalla for providing early feedback on the idea.

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
