# OpenReview forum: "Additive MIL: Intrinsically Interpretable Multiple Instance Learning for Pathology"
_NeurIPS.cc/2022/Conference — NeurIPS 2022 Accept_

### Official Review · Reviewer_8F8M · 2022-06-24

**Rating:** 8
**Confidence:** 4
**Soundness:** 4 excellent
**Presentation:** 3 good
**Contribution:** 3 good

**Summary:**

This paper propose an additive MIL technique to improve interpretability of MIL techniques. The key idea is to make prediction per instance, and summing ("additive") these predictions to get the final probability. This allows the author to get 1) excitatory/inhibitive signal for each patch 2) per-class contribution for each patch 3) a more linear instance contribution and 4) more straightforward way to understand interactive between patches (since their signal are getting added together). The author perform experiment on two TCGA cancer types and Camelyon 16, and demonstrate no loss of performance with respect to the state of the art, while displaying many desirable quality for interpreting the model's prediction.

**Questions:**

- Because of the change in operation order, there seems to be much more computation that needs to happen (prediction for every patch). How does this affect run time of the method compared to SoA?
- Why is the squares in Fig 3 appear to have different sizes in each column?
- It could be helpful to give the reader the sense of scale for each dataset (how many instance per bag, how many bag total, how much of the instance is needed for the bag to be considered positive, etc). Some of these are already covered in text, but I think it could be helpful to see it side-by-side in a table. A problem with 10-instance per bag is going to be different from a problem with 10k-instance per bag (which I think is the case here and making the technique even more significant).

**Limitations:**

Yes

**Strengths And Weaknesses:**

- The method is very simple and appear to have worked well. It consists only change in order of operation, where the prediction of each patch is summed after the final logit, rather than the input.
- There is a theoretical guarantee that the per-instance prediction does represent the marginal prediction.
- Because the prediction are being added together, the comparison/interpretation of the each per-instance prediction is direct, because each prediction contribute linearly to the final prediction.
- The evaluation task is realistic, using sizeable datasets that are publicly available. The problem are varied enough, from micromet detection (which is more aligned with classical MIL problem) to cancer subtype classification (where the judgement may require taking average rather than the max).
- My main concern is whether or not the additive MIL heatmap was really helpful for interpretation. It is true that we can see per-class excitation/inhibition from the heatmap, but it is unclear if these actually are meaningful. The authors claim (such as in Fig 4), that the heatmaps are aligned well with the tissue, but without extensive knowledge of tumor pathology, it is impossible for the reader to judge if the claim is correct or not. I recommend the author obtain pathologist segmentation of each tumor subtypes, so that we can evaluate better if the model heatmap actually align with human expectation.

To this point, the author did look at the problem on Camelyon, but only one small patch is shown on this binary classfication problem.
- Figures can be labeled more clearly. It took me sometime to understand what each color is in Fig 3-6.  One idea is to add colorbar to clearly label every color used. In fig 3, there is only one legen in the top middle figure, when this hsould been to the side/top so it's more clear that this apply to every panel.

---

> ### Author Response · Authors · 2022-08-02
> **Author Response**
>
> Thanks for your extensive comments and thoughtful suggestions! We wanted to take this opportunity to respond to some of your questions:
>
> 1. We agree that the value of the heatmaps and additional interpretability is best evaluated by an expert pathologist. To show that the regions highlighted by a pathologist coincide better with our additive MIL heatmaps, we densely annotated the Camelyon16 slides to conduct the study mentioned in Section 3.3.2. However due to time constraints, we could not get exhaustive annotations for tumor subtype. A significant complication of tumor subtyping is inter-pathologist variability in assigning tissue regions to a subtype, indeed some cases clinically require a specialized immunohistochemistry stain to make this evaluation. Nevertheless, we did attempt to address this in our supplementary submission where in Section 4, Table 1, we conduct a qualitative assessment of the attention and the additive MIL heatmaps with a board-certified expert pathologist. The expert reviewed 50 slides from Camelyon16 (micrometastasis detection) and 39 slides from TCGA RCC (kidney cancer subtyping) and answered which heatmap of the two would be most useful. The pathologist preferred the additive heatmap in 33/39 slides for subtyping and 49/50 for micro metastasis detection. While qualitative we believe this data strongly suggests that the additive heatmaps better align with human expectation and will be more useful in a clinical context.
> 2. We have updated our figures by adding colorbar legends to them. Thanks for this suggestion.
> 3. We’ve also added the details around the number of patches, bags and slides in each dataset in Section 3.1.
> 4. You rightly noted that even though the number of parameters in both the Additive and non-Additive models stay the same, the computation needed for an Additive model is slightly more due to the function being applied to every patch and being summed later. However, this is all still a vectorized matrix multiplication operation on the GPU and does not add a lot of additional computation. Specifically, the additive model had 30.04 GFLOPS as compared to the 29.98 GFLOPS of the non-additive model which resulted in an increase of the forward pass time by 0.08% using a Quadro RTX 8000 (averaged over 100 iterations with a bag size of 100).

---

> > ### Comment · Reviewer_8F8M · 2022-08-09
> > **Post Rebuttal**
> >
> > Thanks for addressing my comments. The additional experiment in the supplement is helpful, and could be worth highlighting in the main text. All of my other concerns are address, and I am keeping my original rating.

---

### Official Review · Reviewer_MCuA · 2022-07-08

**Rating:** 7
**Confidence:** 4
**Soundness:** 3 good
**Presentation:** 3 good
**Contribution:** 3 good

**Summary:**

This paper presents a method to improve interpretability of attention-based MIL models by using an additive predictor function (last layer head of the model). Although general, the method is applied to histological images where MIL is popular due to the weak nature of slide-level labels. The experimental section shows that making the predictor additive does not reduce the performance of MIL models on 3 histology datasets. The interpretability improvements are shown both quantitatively on a dataset of annotated cancer regions and qualitatively on several examples where class-level and excitatory/inhibitory saliency heatmaps are made possible.


**Questions:**

- I am not sure what the point is of the equivalence with Shapley Values. This is not well explained and the theorem proof comes a bit unmotivated.


**Limitations:**

The authors do address some of the limitations of their approach, mostly the fact that the guarantee of not degrading performance through "additivisation" of a MIL method  was evaluated only on 2 methods and 3 datasets and thus may not hold true for any dataset or MIL method.



**Strengths And Weaknesses:**

strengths:
- clarity/quality: the paper is clear and a joy to read.
- significance: the method is simple and can be retrofitted to existing attention-based MIL models such as ABMIL or TransMIL. It is shown to have little impact on accuracy but does provide significantly improved feedback to pathologists, which is key to the acceptance of AI models.
- empirical study: the authors focus on digital pathology and make a compelling case of the benefit of the method using 3 different public datasets spanning different organs and diseases. First the method is shown to have little impact on the accuracy of two different SOT attention-based MIL methods, even showing improved accuracy in the case of ABMIL.
- The authors went through the effort of annotating cancer regions on one dataset in order to quantify the improvement in explainability of additive MIL compared to attention MIL.
- A theoretical effort is made to equate the additive MIL with Shapley sampling values, a game-theoretical framework for interpreting model prediction by decomposing by their feature contributions.


weaknesses:
- originality: the true novelty is somewhat limited. The paper combines two methods: attention-based MIL and additive attribution models. Nevertheless the combination of the two is original.
- The added benefit over attention-MIL is relatively small, as measured quantitatively against pathologist's annotations: from 0.36 to 0.42 AUPRC. Nevertheless, it shows a significant improvement in false positive reduction.

---

> ### Author Response · Authors · 2022-08-02
> **Author Response**
>
> Thanks for your detailed comments! We appreciate that you liked reading the paper and noted the fact that it could be retrofitted to any previous MIL technique. We agree with your assessment that the equivalence of Additive MIL predictions with Shapley values could’ve been better motivated. Shapley values represent the exact marginal contribution of patch towards a prediction, making them ideal for interpretation. However, computing Shapley values requires computing over combinatorial sets which, in practice, is computationally intractable. Therefore, previous methods in interpretability literature use approximations. In our proof we show that our additive MIL formulation is equivalent to Shapley values without approximation, we believe that this represents a significant technical contribution and provides theoretical support for our approach and results. We have re-written Section 2.3 to better highlight why this equivalence is an important property of Additive MIL models in the context of patch attribution. We have also moved the proof to the appendix so it’s not distracting to the reader.

---

> ### Comment · Reviewer_MCuA · 2022-08-08
> **acknowledgement of author's response**
>
> I appreciate the author's response. In particular the clarification of the theoretical analysis with Shapley values. I also think that moving the proof to the appendix is good as the focus of the paper is not on this aspect of the work. Overall, I keep my good rating.

---

### Official Review · Reviewer_cSea · 2022-07-11

**Rating:** 6
**Confidence:** 4
**Soundness:** 3 good
**Presentation:** 3 good
**Contribution:** 3 good

**Summary:**

The authors present a method, inspired by generalized additive models, for performing MIL on digital pathology datasets in order to improve model interpretability. Additive MIL models allow for straightforward patch credit assignment where spatial interpretability is improved over attention MIL methods alone. Additive MIL can be applied to any MIL method that includes a pooling operation.

**Questions:**

Please see weakness above.

**Limitations:**

Yes.

**Strengths And Weaknesses:**

Strengths:
1) The authors provide an excellent explanation for the reasons why attention coefficients in attention-based MIL cannot be interpreted simply as the most positive representatives of given class. I have had a similar thought and the authors articulate this very well.
2) The proposed methods does provide an additional and more informative level of interpretability that is demonstrated well in the figures. The authors are commended for emphasizing this work is about interpretability, not about improved predictive performance.
3) The method is simple and general, which is the mark of a novel and important contribution.

Weaknesses:
1) I did not find the derivation of Theorem 1 (Shapley values) helpful for understanding the importance or major contributions of the paper. This derivation seemed out of place or added to increase the content/technical contribution of paper. I leave this to the other reviewers if they believe it is helpful.
2) I would recommend the authors include results on the use of Additive MIL withOUT an attention module (w/o AB or transformers). I understand that this is not the emphasis of the paper, but it will help disentangle the importance of attention and additive MIL for predictive performance (i.e. is additive MIL sufficient?). These results can be included in the appendix or supplemental data. Decreased predictive performance with Additive MIL only does not weaken the paper.

---

> ### Author Response · Authors · 2022-08-02
> **Author Response**
>
> Thanks for your comments and thoughtful feedback! It’s great that you share our view on how attention coefficients in attention MIL are not straightforward to interpret. To answer the two questions you mentioned in your comment:
>
> 1. We agree that the equivalence of Additive MIL predictions with Shapley values could’ve been better motivated. Shapley values represent the exact marginal contribution of patch towards a prediction, making them ideal for interpretation. However, computing Shapley values requires computing over combinatorial sets which, in practice, is computationally intractable. Therefore, previous methods in interpretability literature use approximations. In our proof we show that our additive MIL formulation is equivalent to Shapley values without approximation, we believe that this represents a significant technical contribution and provides theoretical support for our approach and results. We have re-written Section 2.3 to better highlight why this equivalence is an important property of Additive MIL models in the context of patch attribution. We have also moved the proof to the appendix so it’s not distracting to the reader.
> 2. We implemented the MIL model without attention using a fixed mean pooling function as you suggested. We agree that showing this disentangles the use of attention and the additive formulation. We have updated the paper with these results.

---

### Meta-Review · Area_Chair_MfTV · 2022-08-26

**Recommendation:** Accept
**Confidence:** Certain

**Metareview:**

generalized additive models, for performing MIL on digital pathology datasets in order to improve model interpretability.

The reviewers find that the paper is well written and describes the problem setting as well as their solution well.
The method is considered a valuable way to provide interpretability, is general and novel.
The empirical evaluation shows the benefit on three different data sets.
The reviewers agree on acceptance of the paper.



**Award:**

No

---

### Decision · Program_Chairs · 2022-09-14

Accept